# Morphological, Behavioral, and Molecular Characterization of Avian Schistosomes (Digenea: Schistosomatidae) in the Native Snail *Chilina dombeyana* (Chilinidae) from Southern Chile

**DOI:** 10.3390/pathogens11030332

**Published:** 2022-03-09

**Authors:** Pablo Oyarzún-Ruiz, Richard Thomas, Adriana Santodomingo, Gonzalo Collado, Pamela Muñoz, Lucila Moreno

**Affiliations:** 1Laboratorio de Parásitos y Enfermedades de Fauna Silvestre, Facultad de Ciencias Veterinarias, Universidad de Concepción, Chillán 3780000, Chile; richardthomasshz@gmail.com (R.T.); adrianasantodomingo@gmail.com (A.S.); 2Departamento de Ciencias Básicas, Facultad de Ciencias, Universidad del Biobío, Chillán 3780000, Chile; gcollado@ubiobio.cl; 3Instituto de Patología Animal, Facultad de Ciencias Veterinarias, Universidad Austral de Chile, Valdivia 5090000, Chile; pamela.munoz@uach.cl; 4Departamento de Zoología, Facultad de Ciencias Naturales y Oceanográficas, Universidad de Concepción, Concepción 4030000, Chile; lumoreno@udec.cl

**Keywords:** avian schistosomes, freshwater snails, Chilinidae, cercarial dermatitis, swimmer’s itch, Neotropics, Chile

## Abstract

Avian schistosomes are blood flukes parasitizing aquatic birds and snails, which are responsible for a zoonotic disease known as cercarial dermatitis, a hypersensitive reaction associated to the cutaneous penetration of furcocercariae. Despite its worldwide distribution, its knowledge is fragmentary in the Neotropics, with most of data coming from Argentina and Brazil. In Chile, there are only two mentions of these parasites from birds, and one human outbreak was associated to the genus “*Trichobilharzia*”. However, the identity of such parasites is pending. The aim of this study was to identify the furcocercariae of avian schistosomes from Southern Chile using an integrative approach. Thus, a total of 2283 freshwater snails from different families were collected from three different regions. All snails were stimulated for the shedding of furcocercariae, but only *Chilina dombeyana* (Chilinidae) from the Biobío region was found to be parasitized. The morphology and phylogenetic analyses of 28S and COI genes stated two lineages, different from *Trichobilharzia*, shared with Argentina. This study provides new information on Neotropical schistosomes, highlighting the need for major research on these neglected trematodes, which are considered to be emerging/re-emerging parasites in other parts of the globe as consequence of anthropogenic disturbances and climatic change. Highlights: 1. Two different lineages (Lineage I and II) were described and molecularly characterized (28S and COI genes); 2. Cercaria chilinae I y II are proposed as a synonymous of Lineage II. Thus, a total of four different lineages of avian schistosomes are related to *Chilina* spp.; 3. *Chilina* spp. represents an important intermediate host for avian schistosomes in South America, constituting a reservoir de schistosomes with zoonotic potential; 4. Coinfection between the two different lineages was found, a finding previously not reported for avian schistosomes; 5. Expansion in the geographic distribution of *Nasusbilharzia melancorhypha* from its original record in Argentina, with *Chilina dombeyana* as an additional intermediate host.

## 1. Introduction

Avian schistosomes are parasitic flukes inhabiting the blood stream of aquatic birds of different orders. This group of flukes is composed of 11 described genera, although the genus *Trichobilharzia* is considered to be the most specious taxon with about 40 species [1,2,3,4]. The lifecycle of this group of flukes involves aquatic birds as the definitive hosts, where adult worms will copulate and release their eggs through feces, for visceral schistosomes, or through nasal secretions, in the case of nasal schistosomes [3,5]. Then, the miracidium hatches and looks for its specific intermediate host, an aquatic snail. Inside the gastropod, it transforms to sporocysts, which are intimately associated with the hepatopancreas. After about 3–10 weeks, these sporocysts start to produce furcocercariae, which are released into the water column, where they begin their search for a definitive host and restart the lifecycle [3,5,6].

Notwithstanding the above, these cercariae are capable of penetrating the skin of humans or other mammals. This occurs given the presence of fatty acids in the skin, which are shared with aquatic birds, prompting the accidental cutaneous penetration of these parasites and causing a zoonosis called Swimmer’s itch or cercarial dermatitis (CD) [5,7]. This illness occurs when humans contact waterbodies infested by these larvae. However, during the first contact, there could be a mild cutaneous reaction or the absence of any reaction, although the schistosomula could reach other organs after traveling through the bloodstream [6,7]. When humans have been repeatedly exposed to these cercariae, sensitization occurs, causing a cutaneous reaction, which is characterized by macular eruptions, erythema, and urticaria accompanied by intense itching, which could culminate with pustules if bacterial contamination occurs. This clinical presentation could last for hours or weeks [3,7,8]. Further, vulnerable age groups such children and elderly people seem to develop a more complicated disease course, such as the development of a severe fever [8]. Additionally, some authors suggest, as based on experiments in a murine model, that the cercariae could differentiate into a schistosomula in human skin, migrating to different organs such as the liver, lungs, and even the central nervous system, resulting in a more severe illness [7,9]. Recently, CD has been reported as an emerging disease, both due to the recent outbreaks that have occurred in areas where the parasitosis was unknown, and because the increased number of outbreaks was likely associated with climate change in countries with long-term occurrence of this parasitosis [3,6].

There is a noteworthy correlation between the family of intermediate hosts and the continent on which these schistosomes are distributed. For example, in Europe, most of the snails acting as hosts for avian schistosomes are taxa from the family Lymnaeidae; in North America, this is the family Physidae, followed by Lymnaeidae [2,3,10,11]. However, in South America, there are several taxa that host the larvae of avian schistosomes, such as Ampullariidae, Planorbidae, Physidae, Lymnaeidae, and Cochliopidae, although species of the family Chilinidae are the more frequently identified hosts for schistosomes (see [12]). Nevertheless, in South America, there is a concentration of data on avian schistosomiasis, including worms and human cases in Brazil and Argentina [4,12,13,14,15], while for the remaining South American countries, there are only few mentions or a complete lack of information [4,12,16]. 

The systematics of Neotropical schistosomes is poorly understood both for freshwater and marine schistosomes [4,12,13,14,15,17]. So far, five genera and seven species have been recorded in the Neotropics [4,12], although most correspond to older records with no molecular data. In addition, there are about 10 different furcocercariae whose specific identities remain unknown [12,18,19,20]. Molecular characterization is particularly relevant for the identification of larval stages that have been recovered from the snail hosts, which are hard to classify at the species level. Identification would enable comparisons to be made with available sequences from potential definitive hosts, contributing to the elucidation of their lifecycles [2,3,10,11,21].

In Chile, there is limited information on avian schistosomes, with only two previous records noted in aquatic birds—the first in a Chilean flamingo (*Phoenicopterus chilensis* Molina, 1782) from Northern Chile [22], and the most recent record in a black-necked swan (*Cygnus melancoryphus* (Molina, 1782)) from Southern Chile [16]. Both cases recorded only worms and eggs at histopathological examination. With respect to CD across the country, there is only one previous record of an outbreak from the Laguna Chica de San Pedro, Biobío region in 2006 [23]. The responsible schistosome was identified as “*Trichobilharzia* sp.”; however, considering the morphological features of these furcocercariae and the intermediate host involved (a snail of the family Chilinidae), doubt was recently cast on its identity [13].

Although there are some data on the presence of avian schistosomes in the country, there is no specific information on their biology, ecology, or DNA sequences. Thus, the aim of the present study was to characterize, for the first time, the hosts, behavior, morphology, and molecular data of avian schistosomes distributed in snails from Chile. Further, the aim was also to establish a framework for future studies regarding the biology and systematics of these unknown flukes in the Neotropics.

## 2. Results

### 2.1. Freshwater Snails as Intermediate Hosts

Of all snail species collected, only 35 *C. dombeyana* from Laguna Chica de San Pedro de la Paz, Biobío region (36°50′58.509″ S 73°5′5.133″ W) were parasitized by intramolluscan stages of avian schistosomes, including Schistosomatidae sporocysts and furcocercariae. Including all snail taxa, the total prevalence was 1.53% (35 out 2284 snails). Meanwhile, the prevalence for *C. dombeyana* was 4.9% (35 out 714 *Chilina* snails) and for Laguna Chica de San Pedro the prevalence was 7.83% (35 out 447 *Chilina* snails). Further, of the 35 parasitized *C. dombeyana*, 25 snails released a total of 3893 furcocercariae (_M_I = 155.72; _M_A = 5.45; R = 1–908). The parasitized snails were collected between November 2019 and February 2020. Two lineages of furcocercariae were recognized from *C. dombeyana*, with Lineage I isolated from 5 snails and Lineage II from 22 snails. In addition, three parasitized snails showed co-infections between Lineages I and II.

### 2.2. Morphological Description of Intramolluscan Stages

#### 2.2.1. Sporocysts

These structures were found as an intricate net that was deeply associated with the digestive gland of snails, although when the infections were severe, these sporocysts were also found over other organs such as the crop. In addition, sporocysts showed weak pulsatile behavior, probably associated with the immature cercariae inside them. These sporocysts had a smooth tegument and they were grey-colored, thin-walled, and tubular-shaped with a maximum length of 5 mm. 

#### 2.2.2. Furcocercariae

The fusiform body was covered with small spines, and it had a tail stem of variable length with two furca. Small spines were seen over the surface of the tail stem. Pigmented eye spots were comprised of a few rounded cells and located in the second third of the body. There was a well-developed penetration organ and acetabulum with marked muscular development. The body and tail stem did not have a swimming membrane, while the furcae showed a dorso-ventral natatory membrane without spines, extending slightly caudal to the end of every furca. The tail stem was comprised of fusiform cells and multiple nuclei. The excretory canals opened at the posterior border of each furca. There were five pairs of penetration glands organized as follows: two pairs anterior to the acetabulum, another pair at the acetabulum level, and another two pairs posterior to the acetabulum with the last pair being larger. The apharyngeata cercariae had a short esophagus and two small ceca bifurcating slightly posterior to eye-spot level. All of these traits were shared between the two lineages (Figure 1, Table 1).

Lineage I (Figure 1 and Figure 2). The penetration organ was very well developed with a strong musculature at its basis. The first to fourth penetration glands were composed of larger granules; the last caudal pair was composed by smaller and hyaline granules. The genital primordium was evident only with Alum Carmine stain. This organ was semi-circular with multiple compacted cells and was located immediately to the posterior border of the acetabulum. The body was similar in length to the tail stem; meanwhile, the furcae were about half of the length of the tail stem. The protonephridial system was organized by five pairs of flame cells in the body and the other pair in the tail stem. Those from the body were organized as follows: two pairs between the eye spots and acetabulum; another another pairs of flame cells posterior to the acetabulum; and a final pair in the tail stem, immediately after the point where it joins with the body. The protonephridial formula was 2[(2) + (3) + (1)] = 12. 

Regarding the tegumental traits seen under scanning electron microscopy (SEM), the body spines were concentrated at the cephalic organ and decreased in density posterior to the acetabulum. Further, at the flanks of the body—mid-distance between the acetabulum and posterior border of the body—and next to the ventral area of the posterior border of the body, there were no spines. Some papillae-like structures surrounded the enzymatic apertures of the cephalic organ. The acetabulum was covered with spines, but only around its aperture; these were slightly larger than those from the body. The entire surface of the tail stem was covered by small spines, larger than those on the body. Further, its surface was covered by a delicate honeycomb-shaped mesh, which was found only at the anterior third of its length. The furcae showed densely organized spines on its entire surface, although they were thinner than those of the tail stem (Figure 2).

Lineage II (Figure 1 and Figure 3). The penetration organ had weakly developed musculature at its basis. There were three pairs of papillae that were laterally located on the body: two pairs were located over the anterior border of the penetration organ, and the other pair at the basis of the same organ. The first pair of penetration glands, anterior to the acetabulum, was composed of larger granules, some of which were dark in appearance. Meanwhile, the rest of the penetration glands had smaller and hyaline granules. The genital primordium was evident only with Alum Carmine stain. This organ was oval with multiple compacted cells and was located between the mid-distance of the acetabulum and the posterior border of the body. The length of body was about half the length of the tail stem. The furcae are short when compared to the tail stem, at about a third of its length. The protonephridial system was organized by six pairs of flame cells in the body and the other pair in the tail stem. Those from the body were organized as follows: the first pair was located laterally at some distance from the base of the penetration organ, although in some specimens, it was difficult to find; two pairs were between the eye spots and acetabulum; three pairs of flame cells were posterior to the acetabulum; and the last pair was found in the tail stem, immediately after the point where it joined to the body. The protonephridial formula was 2[(3) + (3) + (1)] = 14. 

Regarding the tegumental traits seen under SEM, the entire body was covered with small spines, although with a marked concentration over the penetration organ. Additionally, several pores were seen in the tegument of the body. Some papillae-like structures were seen around the enzymatic apertures, and some small apertures were seen at the basis of the penetration organ; it is likely that small cilia were attached there. The acetabulum had small spines around its aperture, although they were shorter in comparison to Lineage I. Furthermore, there were a pair of small cuticular patches with spines anterior and posterior to the acetabulum. For the anterior patch, there were two papillae-like structures, one on each vertex, at the basis of the acetabulum. The tail stem was covered on almost its entire surface by spines that are larger than those from the body, although immediately after the point where it joins with the body, there is a small area without spines. In the first quarter of the tail stem, a delicate honeycomb-shaped mesh was seen, which disappeared from the caudal view. From anterior to caudal, the spines of the tail stem had reduced in size and density starting at the second third of its length. The furcae showed larger spines in comparison with those from the last quarter of the tail stem, covering its entire surface (Figure 3).

### 2.3. Release, Behavior, and Life Span of Cercariae

Most furcocercariae emerged during the first hours of the day, between 8 a.m. and 12 p.m., and only a few were collected during the afternoon. After emerging from the snail, the cercariae agglomerated in the illuminated area of the well, suggesting positive phototactic behavior. These cercariae swam by shaking both the body and tail vigorously, and the latter was done in a circular manner, moving forward with the tail and, on some occasions, with the body. After a few seconds (about 10 s), they rested with the ventral side of their bodies extended over the surface of the water, holding their bodies with the aid of the acetabulum. The tail stem was in an approximately 45° angle with respect to the body, and the furca was in a 90° angle in regard to the tail stem, with occasional folding movements of the furca. This resting behavior was brief (about 5 s), then the swimming behavior resumed. In addition, other cercariae were attached to the wall of the well with the help of the acetabulum; this was always associated with the illuminated area, resting for about 1 min, with occasional movements of the body and tail. Meanwhile, a few cercariae were attached to the bottom of the well. After the water surface was disturbed with a metal handle, both the cercariae that were resting on the surface of the water and those adhered to the well resumed their vigorous movements, which could be considered as a mechanically positive behavior. The described behavior was shown by the identified Lineages I and II. 

With respect to the life span of furcocercariae, we suggest that they have a maximum life span of 24 h under the described laboratory conditions. This was determined in the moments spanning from when they emerged from the snail until we found dead cercariae in the bottom of the wells.

### 2.4. Phylogenetic Analyses

The maximum likelihood (ML) and Bayesian inference (BI) for 28S and COI genes supported the morphological diagnosis of the two isolated avian schistosomes, Lineages I and II. These lineages were associated with the other two lineages recorded in Argentina; Lineage I was related to lineage 1 [12] and *Nasusbilharzia melancorhypha* Flores, Viozzi, Casalins, Loker & Brant, 2021 [4], and Lineage II was related to lineage 2 [12]. In both cases, there was high node support for ML and BI. These lineages were closely related to the clade BTGD (Figure 4), the major avian schistosome clade according to Brant & Loker [2]. ML and BI phylogenetic analyses for the COI gene showed that the three known lineages of schistosomes were related to *Chilina*, which was clustered in a big monophyletic clade and separated from the rest of the taxa associated with the BTGD clade, although with mild node support. Lineage II, from the present study, and Lineage 3 from Argentina were closely related, although with mild node support. For the 28S gene, the three lineages were separated into three different clades, each of which was well supported; the phylogenetic analyses for the COI gene further supported these findings. However, Lineage I constituted a well-supported clade, which was basal to most of the species from clade BTGD, except for *Bilharziella polonica* (Kowalewski, 1895; Looss, 1899). On the other hand, avian schistosomes of Lineage II were closely related with lineage 3 from Argentina and other unidentified schistosomes transmitted by different gastropod families (Figure 4); this finding had robust node support. In addition, the ML and BI analyses of the concatenated sequences showed a tree of similar topology to that of 28S; Lineages II and 3 were closely related in a monophyletic clade with robust node support, and Lineage I was basal to nearly all of clade BTGD, except for *B. polonica*, which had robust node support, including the genus *Allobilharzia* (Kolářová, Rudolfová, Hampl & Skírnisson, 2006 [24]), which parasitizes swans (Figure 5).

## 3. Discussion

This study is the first to provide morphological and molecular descriptions of the avian schistosomes present in Chile, including their cercarial behavior. Two different lineages of avian schistosomes from *C. dombeyana* were recorded, each one distinctive from the other in terms of their morphology and morphometry, and they were further supported by molecular data. All infected snails came from the same lagoon, Laguna Chica de San Pedro, where Valdovinos & Balboa [23] reported an outbreak of swimmer’s itch. This outbreak was associated with “*Trichobilharzia* sp.”; however, this taxon was not properly described and did not include all measurements nor molecular characterizations, which are required to determine the identity of furcocercariae [2,10,11,21,25].

Measurements of Lineages I and II were in complete accordance with those from lineage 1 and 2 reported from Argentina. Lineage I, lineage 1, and *Cercaria chilinicola* [20] shared well-developed musculature at the base of the penetration organ. However, the cercariae of Martorelli [20] differed in regard to Lineage I because this cercaria was completely covered with small spines, which was not true for Lineage I. Further, measurements such as body length, tail stem length, furcae length, and the proportions between these structures were different between these taxa. Flores et al. [12] suggested that lineage 2 and *C. chilinicola* would correspond to the same taxon because they shared several morphological traits and measurements. A similar situation was observed with our furcocercariae from Lineage II; however, there were some morphological differences, including the following: the eye spots of *C. chilinicola* were in the first third of body, in comparison with the rest of the schistosomes associated with *Chilina*, including those in our samples, which were found in the second third of the body. In addition, sporocysts of *C. chilinicola* were different between Lineages I and II because the former was rounded in one end and had two rounded tips at the other end, but sporocysts from the isolated lineages were rounded with a single tip at both ends.

In the case of *Cercaria chilinae I* described by Szidat [18], the total length, length of the body, and the proportion between the length of the body and length of the furcae were aligned with Lineage II, although the tail stem was slightly longer, and the proportion of the length of the body and length of the tail stem was slightly smaller. According to Ostrowski de Núñez [26], *C. chilinae* I and II probably correspond to the same taxon since the observed differences were negligible. We support that suggestion, because the measurements for both schistosomes (see Table 1, Szidat [18]) are included in the range of measurements for Lineage II from the present study, with the exception of tail stem length and total length, which were slightly longer. These morphometric differences could be due to the constriction of worms because different fixatives, environmental temperatures, or host size, or they may be a result of their morphometric variability [14,25].

The organization of the penetration glands was the same for the three lineages reported here; however, they differed in regard to the lineages described by Flores et al. [12] from Argentina. According to the former authors, lineage 2 had two pair of preacetabular glands and the remaining three pairs were at the postacetabular location. Meanwhile, for lineage 3, there were two anterior pairs, two pairs at the level of acetabulum, and one pair posterior to it. Besides, *C. chilinicola* [20] had two preacetabular pair of glands, while the other three pairs were postacetabular. However, the worms from the present study have only two pairs of preacetabular glands, one pair paracetabular, and two pairs postacetabular. This organization was found in living cercariae stained with the vital stain Neutral red, a finding that was not reported by the aforementioned authors. The protonephridial formula of lineage 1 [12] was unknown until now; thus, the present study mentions this for the first time. Meanwhile, for Lineage II, the results were different from lineage 2 from Argentina. According to Flores et al. [12], there were 12 flame cells; however, in the present lineage, there was a pair of small flame cells anterior to the ocelli and caudal to the penetration organ, resulting in 14 flame cells in total, with six pairs of flame cells found in the body instead of five pairs. Both the organization of the penetration glands and the protonephridial formula for Lineage II corresponded to the descriptions of *C. chilinae* I and II in *Chilina fluminea* from Argentina [18]. Consequently, and in spite of the small differences in morphometric data, we suggest that they probably belong to the same species.

A structure composed of multiple small cells was visible only in the furcocercariae stained with Alum Carmine. It was located caudal to the acetabulum at variable distance, according to the lineage. Martorelli [20] suggested that it could be the genital primordium. This conglomerate of cells has not been mentioned for any avian schistosome isolated from chilinid snails (see Szidat [18]; Flores et al. [12]), except for *C. chilinicola* [20]. Its variable position denotes a trait that could be of taxonomic importance; however, its presence and location should first be established in the rest of taxa.

The SEM images allowed us to identify the morphological traits that were unknown for these schistosomes, such as the distribution and size of the spines over the body, acetabulum, tail stem, and furcae, and they also stated the presence of a honeycomb-shaped mesh over the surface of the tail stem for Lineages I and II. To the best of our knowledge, none of the mentioned tegumental traits have been mentioned previously for any avian schistosome cercariae from the Neotropics. As such, these findings highlight the importance of SEM in the tegumental characterization of furcocercariae. Additional studies will establish whether these traits are exclusive to schistosomes parasitizing *Chilina* species, or if they are shared with unrelated taxa. With respect to the tegumental spines, there is a conflict with the description provided by Szidat [18], who stated that the body of *C. chilinae* I was smooth except for the cephalic organ. However, for the lineages reported here, and for the rest of the taxa reported from Argentina, there were small spines that were clearly distributed over the body, although is true that these were more densely grouped over the above-mentioned organ. Considering these similarities, we propose that it is necessary to review the material described by this author for this cercaria and Lineage II.

For the taxon reported by Valdovinos & Balboa [23], the total length that was mentioned is included for both lineages described here, with Lineage I as the largest one, and Lineage II with values in between the two mentioned above. Although these authors performed SEM, none of the traits reported here were described in that study.

The description of cercarial behavior is important for the delineation of taxa in the fluke identification process [3,25]. Thus, this study describes, for first time, the behavior and response to different stimuli of these schistosomes shared with Argentina, except for Lineage III. Szidat [18] and Martorelli [20] described that *C. chilinae* I and II, and *C. chilinicola*, respectively, were positively phototactic and after swimming. They rested by adhering to the wall of the flask or at the surface of the water with their tail at an angle of 90°. The furcocercariae described here showed a similar behavior; however, the angle of the tail was lower with a maximum angle of 45° in relation to the body. The lifespan of furcocercariae was about 24 h, as Szidat [18] also suggested, although Martorelli [20] recorded a longer survival of about 32–36 h.

Molecular characterization proved to be an excellent tool for the identification of schistosome species, particularly in the larval stages, allowing for the linkage of different lifecycle stages without the need for experimental studies, which are time consuming and difficult to achieve [3,14,21]. In the present study, two lineages were aligned with two other schistosomes previously recorded from Argentinean *Chilina* [12], supporting the morphological delimitation of the isolated furcocercariae. Furthermore, one of the lineages, Lineage I, aligned with an avian schistosome that was recently described from the black-necked swan in Argentina, *N. melancorhypha* [4], thus expanding its geographic distribution to Chile, where this swan is also distributed [27]. As a result, for future parasitological surveys on waterfowl, one would expect to find this schistosome. Hereafter, and considering the previous records by Szidat [18], Martorelli [20], Flores et al. [4,12], and the present results, a total of four different lineages of avian schistosomes would parasitize snails of the genus *Chilina*, i.e., Lineage I/1, Lineage II/2, Lineage 3, and *C. chilinicola*, three of which have yet to be described at the species level. However, molecular characterization of *C. chilinicola* is pending.

For the 28S tree and the concatenated 28S-COI tree, Lineages II/2 and 3 shared a common clade—with high nodal support—with *Dendritobilharzia*, *Gigantobilharzia*, and other undescribed taxa of avian schistosomes from freshwater and marine ecosystems transmitted by unrelated gastropods, e.g., planorbids and siphonariids, some of which cause occasional local outbreaks of CD (e.g., Brant et al. [28]; Pinto et al. [14]). This contrasts with the *Trichobilharzia* species, which are responsible for most global outbreaks [3]. Furthermore, Lineage I/1 was located at the basal position to almost all taxa of the BTGD clade, including Lineages II/2 and 3, with high nodal support, as suggested by Flores et al. [12] as well. In contrast, the COI tree showed the three lineages related to *Chilina* in a common clade with other undescribed taxa, although with mild support; meanwhile, the clade that included *Dendritobilharzia*, *Gigantobilharzia*, and other undescribed avian schistosomes were basal to the rest of BTGD clade. This COI gene tree had a similar topology to that described by Flores et al. [4].

So far, the specific identity of lineages 2 and 3 from Argentina and Lineage II from this study has not been stated. Considering their position in the phylogenetic trees for both genes, and given the genetic distances between the present lineages and other genera of avian schistosomes, these schistosomes would belong to different species or even genera, as Flores et al. [4] recently confirmed with the description of the new genus *Nasusbilharzia*. As a consequence, different avian species, not necessarily related to the black-necked swan, would host the adult worms of these taxa. Further, other sympatric avian species from different orders, inhabiting the same location of the parasitized snails, could represent potential definitive hosts for these schistosomes, such as coots (Gruiformes), cormorants (Suliformes), herons and egrets (Pelecaniformes), and grebes (Podicipediformes). It is interesting to mention that all these avian orders have been reported as hosts of avian schistosomes in other parts of the world [3].

Although some of these birds have been studied from a parasitological perspective in Chile (see Oyarzún-Ruiz & González-Acuña [29]), all of these studies have been mainly focused on gastrointestinal parasites, so the schistosome fauna remain unknown.

Unexpectedly, a co-infection between cercariae of Lineages I and II was recorded in three snails collected during November 2019. The total intensity of infection varied from 16–85 furcocercariae in these snails; however, only a small part of these cercariae belonged to Lineage I (85 versus 7, 57 versus 1, and 16 versus 1 cercariae). Thus, an evident predominance of Lineage II was stated, which was also true for the rest of parasitized *Chilina*, with Lineage II as the more prevalent taxon. Although there are examples of co-infection between avian schistosomes and other trematodes such strigeids, heterophyids, and echinostomes [5,30,31], to the best of our knowledge, the co-occurrence of avian schistosomes in the same snail host has not been documented in the literature. This finding could be attributed to an immunosuppression of the snails, or it may be that both schistosomes are able to co-inhabit the same host [31]. Furthermore, a degree of competition between both lineages could be suggested, which, according to Lie [30], should be considered as an indirect antagonism between sporocysts, where the dominant taxon causes delayed development in the subordinate one, with a marked reduction or absence of the latter. However, in the present case, the shedding of furcocercariae occurred for both lineages, although in low numbers for one of them, suggesting the presence of a dominant (Lineage II) and a subordinate (Lineage I) taxon. Thus, and following Lie’s definitions, this should be cataloged as weak, indirect antagonism because there was no absolute reduction in the larval output of the subordinate taxon. Notwithstanding the above, even for the dominant lineage, a lower cercarial output was recorded, which was later confirmed through the dissection of these snails. This occurred due to the reduced number of sporocysts, suggesting an early infection. Therefore, to better understand the underlying mechanisms that cause such interactions, experimental studies are required.

According to the previous studies by Flores et al. [12], Szidat [18], and Martorelli [20], other *Chilina* species hosts avian schistosomes, such as *Chilina fluminea* and *Chilina gibbosa*, which are also distributed in Chile [32]. Thus, additional surveys should consider sampling these and other snail species to evaluate if they host the same lineages reported here, or if they host other undescribed lineages. The *Chilina* genus is restricted to the Neotropics; however, some effort has been made to determine the systematics of these snails [12,33], which is particularly important considering that there are 30 morphologically described species in Chile, most of which are concentrated in Central and Southern Chile [32,34]. Hereafter, the molecular characterization of these intermediate host would allow one to establish, for instance, the degree of specificity by these avian schistosomes within the family Chilinidae.

*Physella acuta* is a North American snail that can tolerate marked changes in the abiotic components of the aquatic environment, an advantage that allows them to be a successful invasive species at global scale, including in Chile [34,35]. Furthermore, this mollusk species acts as an intermediate host for zoonotic avian schistosomes in North America [2,11] and recently in Europe [25]. Considering the reduced prevalence rates associated with these parasites (e.g., Horák & Kolářová [6]; Marszewska et al. [8]; Brant & Loker [11]; Al-Jubury et al. [36]), the absence of parasitism by schistosomes in the remaining snail species studied here, such as *P. acuta*, could be related to the small sample size per area, per snail species (in some cases), or it could even be considered an expected result. However, *P. acuta* yielded negative results despite its large sample size when compared with *C. dombeyana*, as there were 1390 versus 714 snails, respectively. This could be related to other biotic factors, such as the presence of dense populations of definitive hosts in the sampled area (e.g., the breeding area; Ebbs et al. [15]; Helmer et al. [25]), or that this physid snail requires migratory waterfowl as a source of infection [13].

During the sampling period, there were no reports of any outbreak of CD, which may be explained by the important distance (about 800 m) between the site where bathing activities occurred and the shore, where parasitized snails were collected. Notwithstanding the above, Szidat [19], Cort et al. [37] and Sckrabulis et al. [38] have reported that furcocercariae can be transported by water currents to relatively large distances, from shore to shore, causing the cutaneous ailment to move away from where the parasitized snails are located. Thus, the risk of CD must not be discarded despite the absence of parasitized snails in the immediate area of bathing activities. Furthermore, according to the lifeguards from the lagoon, Laguna Chica de San Pedro, every summer some bathers appear with cutaneous eruptions after bathing, although they associated it to other causes, such as bacteria or toxins. This could be proof of the current transmission of these parasites after 17 years since its first outbreak in the same location [23].

Another pending issue is related to the zoonotic potential of the isolated lineages. Although Valdovinos & Balboa [23] associated the CD outbreak in Laguna Chica de San Pedro, Chile, to “*Trichobilharzia* sp.”, the present results illustrate that this taxon was, in fact, two different taxa. Thus, experimental studies are required to establish which of these taxa could cause swimmer’s itch, or even if all of them are capable of causing such illness. The sole report of two different lineages represents an important finding and serves as a warning for the potential emergence of a parasitic disease associated with human activities [6,14]. Further, another important issue is that the present Lineage I corresponded to the nasal schistosome, *N. melancorhypha* [4]. Although there are no data on the zoonotic potential of the latter species, these groups of parasites, with *Trichobilharzia 17 egent* studied by Horák, Kolářová & Dvořák, 1998 as the model species, are of concern because in experimental murine models, they are capable of migrating to the nervous system, causing severe pathological changes in the peripheral and central nervous system of mammals. These effects can include paralysis and locomotory disfunction, alterations that can be extrapolated to humans [7,8,9,39].

The likely implications of climate change over parasites have been suggested through shifts in the hosts and geographical range, accelerated development, and increased pathogenicity to hosts. For trematodes, the increase in water temperature would accelerate the development rates inside snail hosts, causing the emergence/re-emergence of these parasites [40]. Consequently, for zoonotic schistosomes, this would translate into an exacerbated transmission to humans, causing outbreaks of CD [3,36]. The current climatic scenario is of concern, highlighting the need for additional research on the early identification of potential zoonotic schistosomes in understudied areas, such as the Neotropics. Furthermore, environmental disturbances related to anthropic activities, such as artificial water bodies and eutrophication, play an additional role in this emergence [5,8,23,41]. This contamination process promotes the excessive growth of aquatic vegetation, which serves as shelter and feeding sites for snails, with subsequent increases in the snail population [5,6], which is what took place at the lagoon Laguna Chica de San Pedro. In addition, modified ecosystems attract invertebrates, such as mollusks and waterfowl, increasing the possibility with which the lifecycles of parasites such as avian schistosomes develop [3,5,8].

Future studies with larger sample sizes per snail species, and with broader geographic distributions, could reveal the occurrence of more schistosome taxa, and can even lead to the discovery of new species. In Southern Chile, there are several lakes, lagoons, and wetlands that house a rich biodiversity of aquatic snails [34] and are usually used for recreational activities; thus, these areas could be at risk for outbreaks of CD, which is of interest for research [3].

## 4. Material and Methods

### 4.1. Sampling and Study Area

A total of 2283 freshwater snails from different families and locations in three regions from Southern Chile were collected between May 2019 and February 2020. These three regions were included in this study because every one of them has an association with CD or the record of avian schistosomes: the Biobío region had an outbreak of CD [23], in the Los Ríos region swans were recorded as hosts of avian schistosomes [16], and the Ñuble region has an unpublished record of CD. The chosen collection sites in every region were characterized by abundant aquatic vegetation, slow-moving waters that guarantee the development of freshwater snails [18,26], and the presence of aquatic birds, thus allowing for the potential fulfillment of the lifecycle of these parasites. Thus, ponds, river borders, lagoons, flooded areas, and wetlands were chosen for the collection of snails. Snails were collected by hand or using a metal net at a maximum depth of 1.5 m [12]. The regions and every location that was included in the present study are detailed in Figure 6. All snail species were identified by a malacologist (G. Collado). For details of the snails sampled in every locality across every region, see Table 2.

### 4.2. Collection and Description of Intramolluscan Stages

Snails were individually arranged in cell culture wells under artificial light for a period of 12 h for 3 consecutive days at room temperature (15–20 °C) to stimulate cercarial emergence. Every snail was checked daily under stereomicroscope early in the morning (8 am), at noon, and in the afternoon (6 pm). Once the furcocercous cercariae were detected, a pool of these were isolated using a glass pipette, stained separately with Neutral red solution (Santacruz), and observed under light microscope for its morphological characterization and to determine its protonephridial organization and distribution and number of penetration glands [2,26]. Released furcocercariae were checked under the naked eye and stereomicroscope to identify the natatorial behavior of every identified lineage. Additionally, the behavioral responses to mechanical and light stimuli were tested. On the third day, all snails were dissected under stereomicroscope to examine for the presence of sporocysts, especially for negative snails, considering that any parasitological descriptor based only on snails releasing cercariae could underestimate the real values for these descriptors [3,21]. All collected sporocysts and cercariae during dissection were preserved in absolute ethanol for molecular analyses.

To measure the structures, a pool of cercariae was preserved in 80% ethanol for subsequent staining with Alum Carmine stain following the methods of Lutz et al. [42]. When co-infection was stated, furcocercariae were sorted under the stereomicroscope according to its general morphology, particularly the evident difference in the length and width of body, and the relation of body length with the tail length. Measurements were stated using the Motic Images Plus 2.0 software associated with the light microscope MOTIC. Further, another pool was preserved in absolute ethanol under freezing conditions at −20 °C for molecular analyses, and another pool was preserved in 80% ethanol for SEM analysis using HITACHI SU 3500 equipment [2,21,43]. Cercariae were disposed in an ionic solution for a few hours (maximum: 4 h) to achieve the appropriate conduction of electrons in the SEM equipment. Given the equipment specifications, there was no need for the gold bathing of worms. Once the flukes were retrieved from the ionic solution, these were disposed over a carbon fiber stub (15 mm in diameter). Then, this stub was placed inside the cool stage of the equipment where they were frozen at −30 °C with vacuum sealing. For every worm, the configuration for the spot values, VP-SEM, Kv, and Pa, was stated.

The parasitological descriptors of prevalence (P), mean intensity (_M_I), mean abundance (_M_A), and range I were reported following the approach detailed by Bush et al. [44].

The furcocercariae and shells of the parasitized snails from the present study were deposited in the collection of the Museo de Zoología, Universidad de Concepción, under the following access numbers: MZUC_UCCC 46707-46710 for snails, and MZUC_UCCC 46711-46714 for furcocercariae.

### 4.3. Molecular Analyses

#### 4.3.1. DNA Extraction and Polymerase Chain Reaction (PCR)

Genomic DNA was extracted by employing two different kits. For the released cercariae, the DNeasy blood and animal tissue kit (Qiagen, Hilden, Germany) was used. In the case of co-infection, the DNA was extracted only from the furcocercariae once these were sorted under the stereomicroscope. Considering that sporocysts are deeply inserted in the snail tissue and given the high presence of mucopolysaccharides in the mollusk tissue, which act as PCR inhibitors, the E.Z.N.A. Mollusc DNA kit (Omega Bio-Tek Inc., Norcross, GA, USA) was used to avoid this potential problem. For both cases, the manufacturer’s instructions were followed. DNA quantity and quality for each extracted sample was measured with an Epoch^TM^ Microplate Spectrophotometer. Samples with values between 1.6 and 2.0 for the A260/A280 absorbance ratio were considered to be pure and suitable for PCR amplification [45]. Finally, DNA was preserved at −24 °C until analysis.

Touchdown PCR was performed to amplify partial sequences of the cytochrome *c* oxidase subunit I gene (hereafter referred to as COI), and of the gene 28S rDNA (hereafter referred to as 28S) [2,13]. Primers and PCR thermal conditions in this study are provided in Table 3. Each PCR reaction was performed following the methods by Dvořák et al. [1] and Horák et al. [21]. Specifically, 3 µL of template DNA was added into a mix of 0.3 µL DreamTaq Polymerase (Thermo Fisher Scientific, Waltham, MA, USA), 0.5 µL dNTPs (0.2 mM), 2.5 µL DreamTaq Buffer, 1 µL of each primer (10 pmol), and 16.7 µL of ultra-pure water to achieve a final volume of 25 µL. Amplicons were submitted to electrophoresis into 2% agarose gels with GelRed^®^ stain (Biotum, Tehran, Iran), and visualized in an ENDURO^TM^ GDS UV transilluminator. Amplicons of expected size were purified and sequenced in both directions at Macrogen (Seoul, South Korea).

#### 4.3.2. Phylogenetic Analyses

The resulting sequences were quality verified and edited with Geneious Prime^®^ version (v) 2021.2.2 (https://www.geneious.com accessed on 4 November 2021) to obtain the consensus sequences. In addition, for comparison, basic local alignment searches were performed with the BLASTn tool (https://blast.ncbi.nlm.nih.gov accessed on 5 November 2021), and similar sequences were downloaded from GenBank (https://www.ncbi.nlm.nih.gov accessed on 5 November 2021) (Appendix A) to build multiple alignments in Geneious, employing the MAFFT algorithm [49].

For both genes, phylogenetic reconstructions were performed using the ML and BI methods with IQ-Tree v1.6.12 [50] and MrBayes v3.2.6 [51], respectively. The best nucleotide substitution models for ML analyses were selected using the ModelFinder algorithm commands “-m MFP+MERGE” for COI and “-m MFP” for 28S [52]. Finally, to evaluate the robustness of the phylogenetic tree inferred with 1000 ultrafast bootstrapping pseudo-replicates, we employed the rapid-hill-climbing and stochastic disturbance methods with IQ-TREE. Values under 70 obtained with ultrafast bootstrapping were considered non-significant, values between 70 and 94 represented mild support, and values over 95 provided robust statistical support [53].

In the BI analyses, we used the following commands to select the best evolutionary models: for the COI dataset that was partitioned into three partitions (codon1, codon2, and codon3), the command “nst = mixed rates = invgamma ngammacat = 4” was used; and for the 28S dataset, the comma“d “nst = mixed rates = gamma ngammaca” = 4” was employed [51,54]. All parameters were unlinked between partitions. Two independent tests of 5 × 10^6^ generations and four Markov chain Monte Carlo (MCMC) chains were run simultaneously, where trees were sampled every 1000 generations, and the first 25% were removed as burn-in. Bayesian posterior probabilities (BPP) with values ≥0.70 were considered as providing strong statistical support [55].

On the other hand, sequences analyzed for both genes through ML and BI were concatenated. These concatenated sequence datasets were created with Mesquite v3.70 [56]. In the case of BI, four partitions were stated: one for 28S (nst = mixed rates = gamma ngammacat = 4); and three others for COI, namely codon1, codon2, and codon3 (nst = mixed rates = invgamma ngammacat = 4). All parameters between partitions were unlinked. For this dataset, two independent tests of 8 × 10^6^ generations and four MCMC chains were run simultaneously, with tree sampling every 1000 generations and with removal of the first 25% as burn-in. In all cases, Tracer v1.7.1. was employed to confirm the correlation and effective sample size (ESS) of the Markov chains [57].

The genetic distances between the isolated lineages in the present study and those used in the phylogenetic analyses were estimated using MEGA7 [58] (see Appendix A). The sequences obtained in the present study were deposited in the NCBI GenBank database under the following access numbers: OM307633-OM307648 for the 28S gene and OM321410-OM321428 for the COI gene (see Appendix A).

## Figures and Tables

**Figure 1 pathogens-11-00332-f001:**
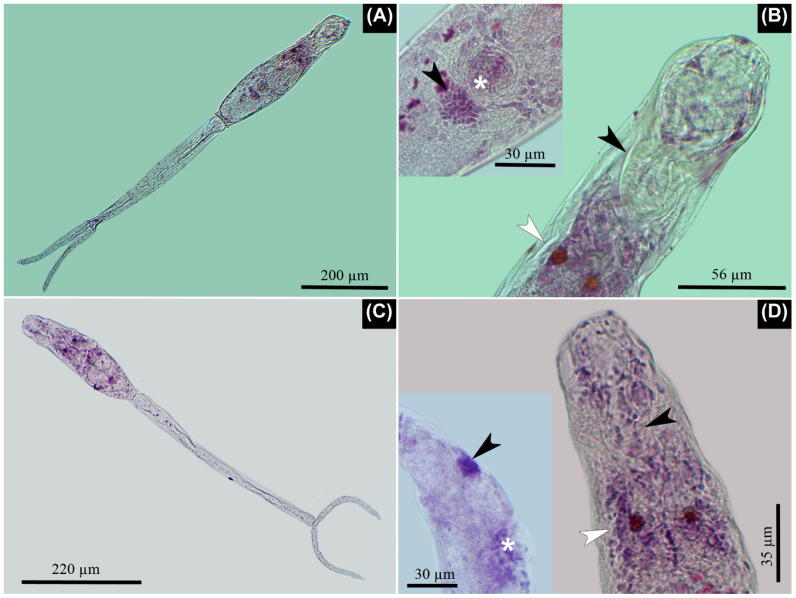
Microscopic images of the isolated lineages stained with Alum Carmine. (**A**,**B**) Lineage I. (**A**) Furcocercaria *in toto*, note the similar length of body and tail stem. (**B**) Well-developed penetration organ in the anterior third of the body with its prominent musculature, which is evident at its base (arrowhead). Note the pigmented eye spots (white arrowhead) posterior to penetration organ. Inserted image: Note the genital primordium (arrowhead) located almost immediately to the posterior border of the acetabulum (asterisk). (**C**,**D**) Lineage II. (**C**) Furcocercaria *in toto*, note the difference between the length of body and tail stem, which is almost two-fold. (**D**) Anterior third of body where the penetration organ is evident (arrowhead); however, its muscular development is less when compared with Lineage I. Note the pigmented eye spots (white arrowhead) posterior to penetration organ. Inserted image: Note the evident distance between the acetabulum (asterisk) and the genital primordium (arrowhead), which is situated caudal to the latter, near the posterior end of body.

**Figure 2 pathogens-11-00332-f002:**
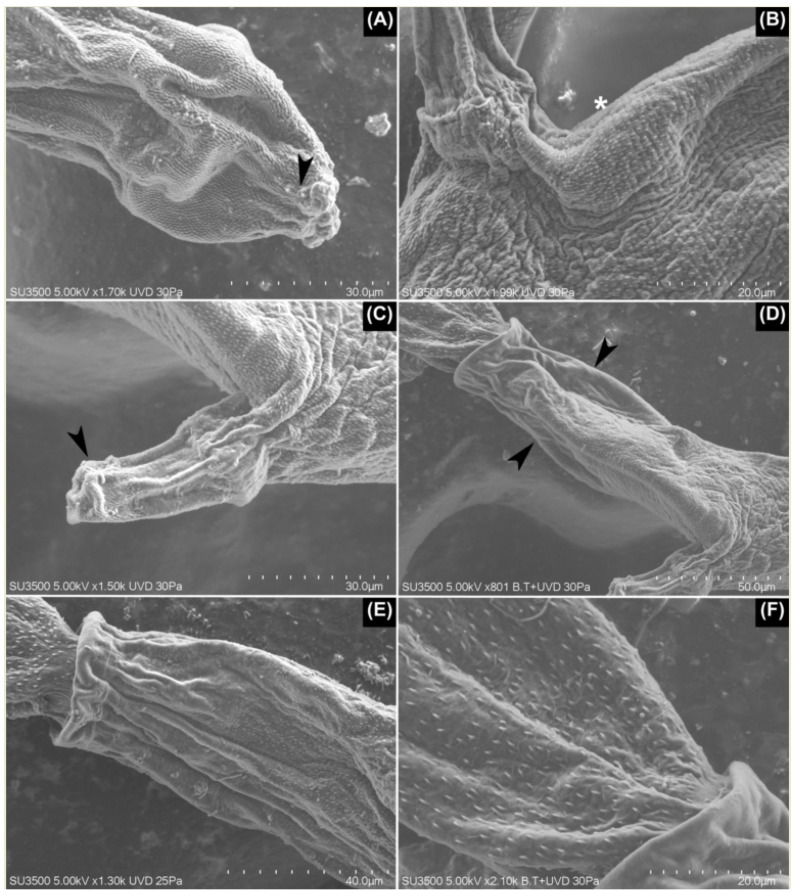
SEM images of Lineage I. (**A**) The penetration organ with a pair of papilla-like structures on the border of the organ (arrowhead). (**B**) The body cuticle with a porous appearance (asterisk), clearly seen posterior to the acetabulum. (**C**) The acetabulum is covered on its tip with small, densely grouped spines (arrowhead), which contrasts with the rest of the acetabulum where no spines were seen. (**D**,**E**) Posterior third of the body where spines gradually disappear (arrowheads) until they are not seen on the posterior border. (**E**) Additionally, note the larger spines of the tail stem in comparison with those from the body. (**F**) The tail stem with a delicate honeycomb-shaped mesh over its tegument.

**Figure 3 pathogens-11-00332-f003:**
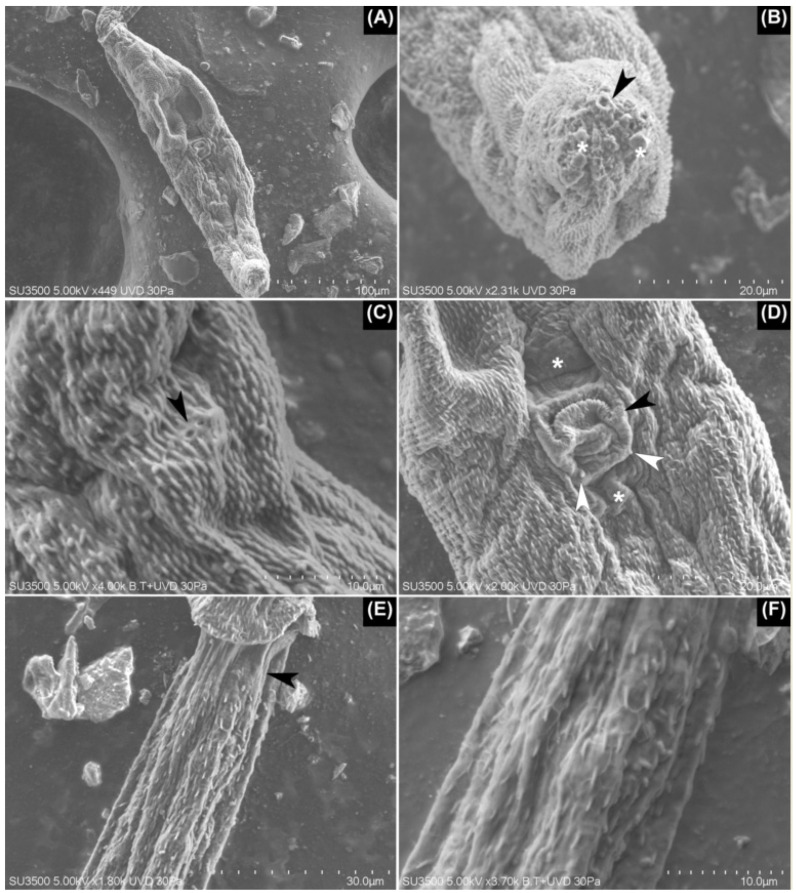
SEM images of Lineage II. (**A**) Ventral view of the body of furcocercariae completely covered with small spines over its tegument. (**B**) Apical view of the penetration organ with its glandular apertures (arrowhead) and some glandular content over these (asterisks). (**C**) The basal portion of the penetration organ where these two apertures were seen likely bore small cilia (arrowhead). (**D**) Inverted acetabulum with some small spines over its border (black arrowhead). Note the presence of two patches lacking spines—one anterior and the other posterior to the acetabulum (asterisks). At the basis of acetabulum, a pair of papillae-like structures are seen (white arrowheads). (**E**) At the anterior third of the tail stem, a small area lacking spines is evident (arrowhead). (**F**) Detail of the delicate, honeycomb-shaped mesh over the tegument of the tail stem, which is also seen in (**E**).

**Figure 4 pathogens-11-00332-f004:**
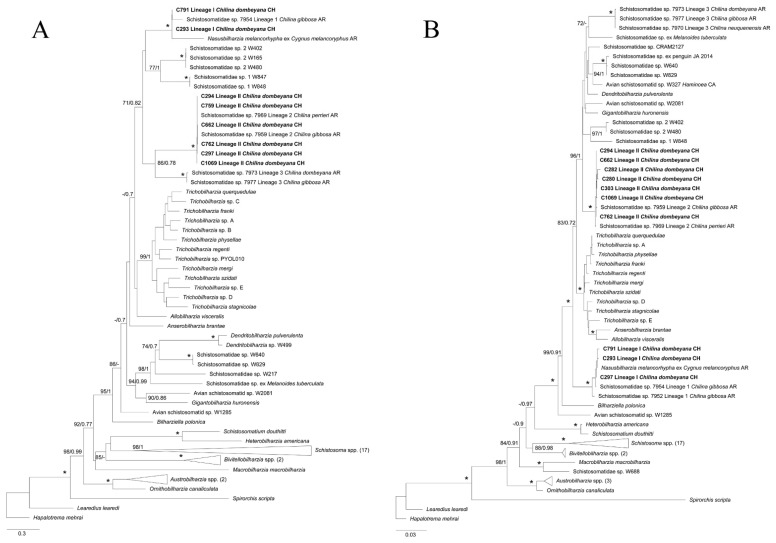
Maximum likelihood (ML) phylogenetic tree for a subset of schistosomes sequences for (**A**) 28S and (**B**) COI genes. (**A**) This phylogeny was inferred using an alignment of 1699 bp. Calculated substitution models for ML and BI were GTR+F+G4, and *M*_203_, *M*_198_, and *M*_200_, respectively. The best models were chosen using the Bayesian Information Criterion (BIC). (**B**) This phylogeny was inferred using an alignment of 1188 bp. Calculated substitution models for ML and BI were as follows: TIM2+F+I+G4 (part1), GTR+F+I+G4 (part2), and TIM2+F+I+G4 (part3); *M*_201_, *M*_138_, *M*_162_, *M*_189_, *M*_203_, *M*_134_, and *M*_193_ (part1); *M*_29_, *M*_92_, *M*_68_, *M*_71_, *M*_81_, *M*_54_, and *M*_180_ (part2); and *M*_125_, *M*_191_, *M*_193_, *M*_200_, *M*_189_, *M*_203_, *M*_166_, and *M*_64_ (part3), respectively. The best models were chosen using the Bayesian Information Criterion (BIC). (**A**,**B**) Bootstrap values ≥ 70 (left) and posterior probabilities ≥ 0.7 (right) are presented at every node. An asterisk (*) indicates full support (100/1). The generated sequences in the present study are highlighted in bold. Outgroups with more than two sequences were collapsed with the number of sequences detailed between parentheses.

**Figure 5 pathogens-11-00332-f005:**
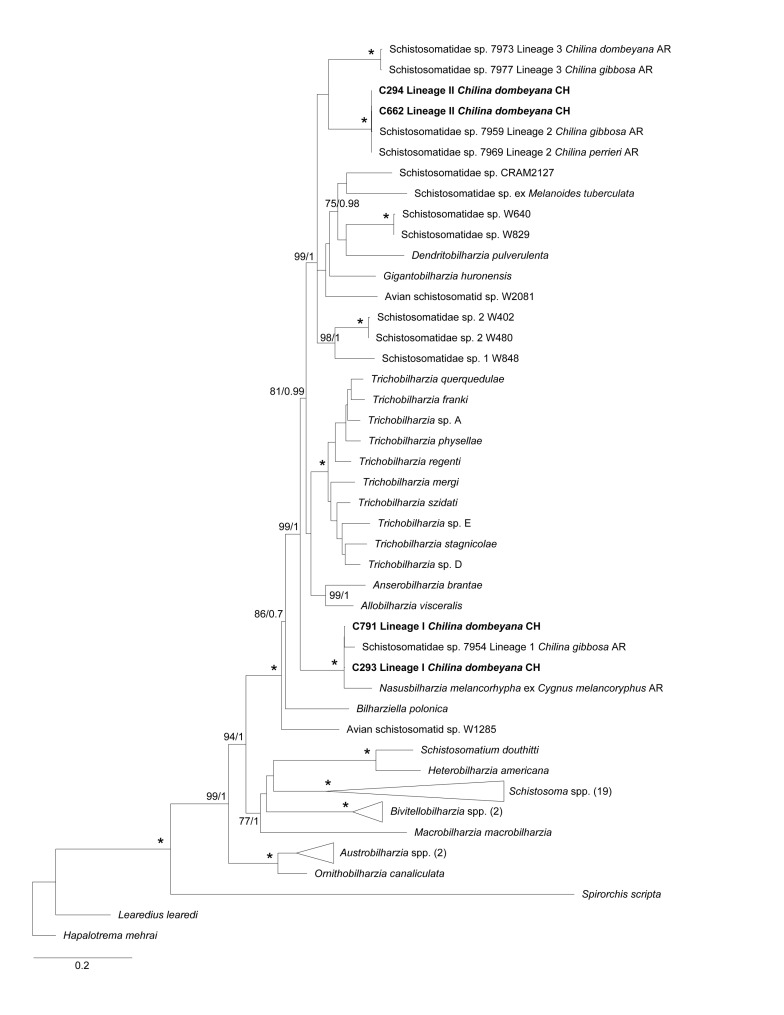
Maximum likelihood (ML) phylogenetic tree for a subset of schistosomes sequences for 28S-COI concatenated genes. This phylogeny was inferred using an alignment of 2894 bp. Calculated substitution models for ML and BI were as follows: GTR+F+I+G4 (nonCoding), TIM2+F+I+G4 (part1), GTR+F+I+G4 (part2), and TIM2+F+I+G4 (part3); *M*_177_, *M*_198_, *M*_195_, and *M*_203_ (nonCoding); *M*_138_, *M*_201_, *M*_162_, *M*_189_, *M*_203_, *M*_193_, and *M*_134_ (part1); *M*_29_, *M*_54_, *M*_68_, *M*_81_, *M*_145_, and *M*_160_ (part2); and *M*_166_, *M*_191_, *M*_125_, *M*_200_, and *M*_203_ (part3), respectively. The best models were chosen using the Bayesian Information Criterion (BIC). Bootstrap values ≥ 70 (left) and posterior probabilities ≥ 0.7 (right) are presented at every node. An asterisk (*) indicates full support (100/1). The generated sequences in the present study are highlighted in bold. Outgroups with more than two sequences were collapsed with the number of sequences detailed between parentheses.

**Figure 6 pathogens-11-00332-f006:**
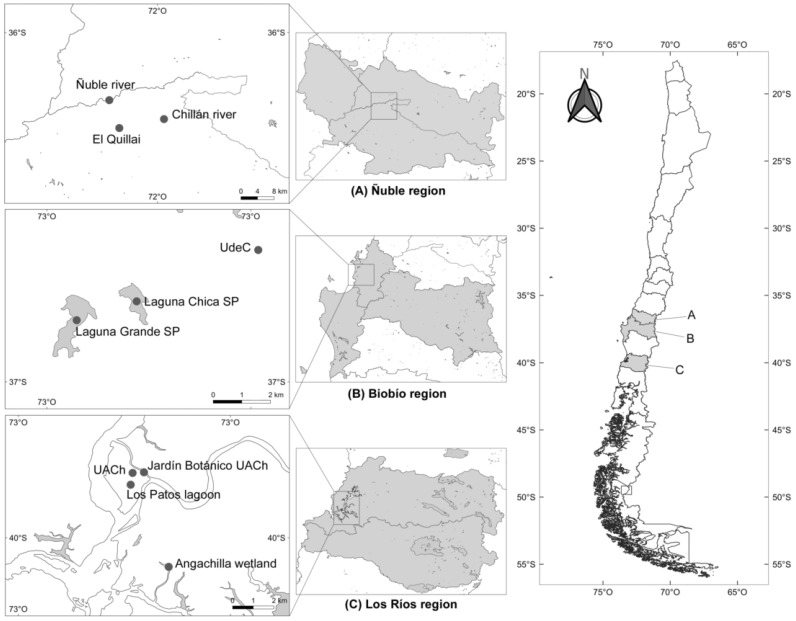
Map of the regions considered in the present study, specifying the sampled locations.

**Table 1 pathogens-11-00332-t001:** Measurements of previously recorded furcocercariae from *Chilina* species, including data from the present study. Adapted from Flores et al. [12]. All measurements are expressed in µm. Values are expressed as a range, followed by mean and standard deviation between parentheses.

Taxon	Lineage I	Lineage II	Species inquirenda *	*C. chilinae* I	*C. chilinae* II	*C. chilinicola*	Lineage 1	Lineage 2	Lineage 2	Lineage 2	Lineage 3
Host	*C. dombeyana*	*C. dombeyana*	*C. dombeyana*	*C. fluminea*	*C. fluminea*	*C. gibbosa*	*C. gibbosa*	*C. gibbosa*	*C. perrieri*	*Chilina* sp.	*C. dombeyana*, *C. neuquenensis*
Locality (country)	Laguna Chica de San Pedro (CHI)	Laguna Chica de San Pedro (CHI)	Laguna Chica de San Pedro (CHI)	Delta Paraná (AR)	Delta Paraná (AR)	Pellegrini Reservoir (AR)	Pellegrini Reservoir, Nahuel Huapi lake (AR)	Pellegrini Reservoir (AR)	Santa Cruz river (AR)	Larga Larga (AR)	Mascardi lake (AR)
Reference	This study	This study	[a]	[b]	[c]	[c]	[d]	[d]	[d]	[d]	[d]
n cercariae	23	96	-	-	-	-	-	10	25	5	20
L total	959.4–1062 (101,006 ± 31.31)	750–937 (839.07 ± 41.98)	684–1212	930	990	1010	1045–1140	805–875	998–1085	1056–1114	816–931
L body	342–432 (378.24 ± 22.12)	185–315 (262.45 ± 24.36)	-	280	280	330	400–435	245–270	259–317	259–278	269–307
W body	61–92 (79.59 ± 8.92)	45–84 (63.92 ± 7.35)	-	70	70	110	90–100	60–65	58–77	67–77	83
L tail stem	354–467 (422.21 ± 29.04)	358–488 (422.55 ± 26.35)	-	650	530	510	410–450	405–450	528–576	595–624	582–624
L furca	193–228 (209.61 ± 9.39)	131–176 (155.47 ± 9.28)	-	-	180	170	235–290	125–175	163–211	182–211	192–240
L × W penetration organ	109–135 (119.82 ± 7.31) × 40–62 (51.59 ± 5.07)	54–93 (73.55 ± 6.07) × 28–52 (39.19 ± 4.26)	-	-	-	90 × 50	100–113 × 50–63	65–85 × 38–45	77 × 40	96–108 × 36–43	99 × 48
D eye spots	7–11 (9.18 ± 1.05)	6–10 (8.22 ± 0.96)	-	-	-	-	-	-	-	-	-
Eye spots position	2nd third	2nd third	-	-	-	1st third	2nd third	2nd third	2nd third	2nd third	2nd third
Flame cells	12	14	-	14	-	14	-	-	12	-	12
D Ace	22–39 (31.86 ± 3.83)	20–36 (26.31 ± 3.01)	-	-	25	35	35–40	20–28	24–29	24–34	29–38
Ae-Ace	199–270 (234.95 ± 19.58)	91–188 (143.69 ± 17.13)	-	200	200	180	250–300	130–150	153	182–202	193
L body: L tail stem	1: 0.76–1.1 (0.9 ± 0.1)	1:0.46–0.78 (0.62 ± 0.07)	-	1:0.43	1:0.52	1:0.65	1:0.9–1	1:0.5–0.6	1:0.5–0.6	1:0.4–0.5	1:0.4–0.6
L tail stem: L furca	1:1.65–2.19 (2.02 ± 0.14)	1:2.32–3.14 (2.72 ± 0.17)	-	-	1:2.9	1:3	1:1.5–1.8	1:2.3–3.4	1:2.5–3.4	1:3–3.3	1:2.4–3.3
L prim	15–29 (21.87 ± 3.49)	15–34 (21.25 ± 3.67)	-	-	-	-	-	-	-	-	-
Ace-prim	0–12 (4.87 ± 2.94)	13–46 (32.73 ± 6.95)	-	-	-	-	-	-	-	-	-

Symbols and abbreviations: * = cited as “*Trichobilharzia* sp.”; AR = Argentina; CHI = Chile; L = length; W = width; D = diameter; Ae = anterior end; Ace = acetabulum; prim = genital primordium; “-“ = information was not detailed. References: [a] Valdovinos & Balboa (2008), [b] Szidat (1951), [c] Martorelli (1984), [d] Flores et al. (2015).

**Table 2 pathogens-11-00332-t002:** Localities considered in the present study, including the sample size per locality for every snail species.

Locality	Coordinates	Snail Species (*n*)				
		*Physella acuta* (Physellidae)	*Chilina dombeyana* (Chilinidae)	*Lymnaea* sp. (Lymnaeidae)	*Potamolithus* spp. (Tateidae)	*Ancylus* sp. (Planorbidae)
**Ñuble region**						
El Quillai	36°39′31.22″ S, 72°12′11.76″ W	20	0	0	0	0
Chillán river	36°38′4.00″ S, 72°4′56.54″ W	1	0	0	0	0
Ñuble river	36°34′59.40″ S, 72°13′49.34″ W	330	8	0	0	0
**Biobío region**						
University of Concepción (Udec)	36°49′41.75″ S, 73°2′13.45″ W	0	25	0	0	0
Laguna Chica de San Pedro (SP)	36°50′54.34″ S, 73°5′5.16″ W	57	447	56	57	0
Laguna Grande de San Pedro (SP)	36°51′21.04″ S, 73°6′29.85″ W	762	9	0	0	24
**Los Ríos region**						
Angachilla wetland	39°51′23.21″ S, 73°14′5.71″ W	93	121	0	5	0
Jardín Botánico (Valdivia city)	39°48′10.46″ S, 73°14′56.07″ W	6	104	0	14	0
Los Patos lagoon (Valdivia city)	39°48′35.85″ S, 73°15′23.13″ W	10	0	0	5	0
Austral University of Chile (UACh)	39°48′11.76″ S, 73°15′19.21″ W	111	0	19	0	0

**Table 3 pathogens-11-00332-t003:** Primers used in the present study, detailing its parameters used in the PCR protocol and the expected size of band for the PCR product.

Gene	Primer	Sequence	To (°C) †	Expected Length (bp)	Reference
COI	Cox1_schis’_5′	TCTTTRGATCATAAGCG	* 50–46 phase 1; 45 phase 2	1000	Lockyer et al. [46]
	Cox1_schis’_3′	TAATGCATMGGAAAAAAACA			
	CO1F15	TTTNTYTCTTTRGATCATAAGC	* 50–46 phase 1; 45 phase 2	600	Brant & Loker [2]
	CO1R15	TGAGCWAYHACAAAYCAHGTATC			
	CO1RH3R internal	TAAACCTCAGGATGCCCAAAAAA			
28S	U178	GCACCCGCTGAAYTTAAG	* 55–51 phase 1; 50 phase 2	1500	Lockyer et al. [46], Tkach et al. [47], Olson et al. [48]
	L1642	CCAGCGCCATCCATTTTCA			
	DIG12 internal	AAGCATATCACTAAGCGG			
	ECD2 internal	CTTGGTCCGTGTTTCAAGACGGG			

† Observable temperature; * Touchdown PCR. The annealing temperature was programmed to reduce to one degree per cycle (50–46 °C; 55–51 °C) for 15 cycles in phase one and a total of 20 cycles in phase two.

## Data Availability

Sequence data, including those generated in the present study as those previously published, are available online at GenBank (https://www.ncbi.nlm.nih.gov/genbank/ accessed on 20 January 2022) in the NCBI Database. Detailed information including the name of every taxon, host, locality, and access code in GenBank for every gene is given in Appendix A. The sequences obtained during the present study are highlighted in bold.

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
