# Peer review of "Morphological, Behavioral, and Molecular Characterization of Avian Schistosomes (Digenea: Schistosomatidae) in the Native Snail Chilina dombeyana (Chilinidae) from Southern Chile"

_pathogens, 2022, doi:10.3390/pathogens11030332_

Round 1
Reviewer 1 Report
This manuscript characterizes the 2 (and in part one more) new in Chile discovered bird schistosome cercariae in a comprehensive approach (including also characteristics such as behaviour, chaetotaxy, genital primordium, protonephridial formula, interaction of the 2 species occurring in the same snail[!]). It discusses the biological details considering the relevant literature in a manner, that the work can be used as a stimulus and guideline for further work.
Author Response
No changes were suggested by the reviewer 1. However, the situation with lineage III was solved, and following recommendations of reviewers it was deleted from the manuscript. Thank your for your words and support, Reviewer 1.
Reviewer 2 Report
First of all, I want to congratulate the authors because they provided a manuscript that presents together morphological, behavioral and molecular results, which is rare nowadays.
I have a main point to raise : lineage III needs to be deleted from the manuscript since not enough data could be provided for it.
The authors will find directly in the manuscript all my concerns in order to improve it and to provide a revised manuscript which will be suitable.

Author Response
REVIEWER 2
page 3, lines 108-109: "explain these numbers. Add 35/447=7.83% at the Laguna Chica"
Response: The number 714 refers to the total snails of species Chilina dombeyana collected from the three regions. Correction was made and information was added. The parasitological descriptors were stated in the material and methods section.
page 3, lines 111-112: "on which basis ? explain the lineages here"
Response: The description of furcocercariae was detailed subsequently. To avoid any confusion, the descriptions of lineages were disposed below the mentioned paragraph.
page 3, lines 131-132: "Lineages were not explained Lineage III only one cercaria and cannot be used to make a third lineage"
Response: Same situation explained above in relation with the descriptions of lineages. Lineage III will be deleted from the manuscript, including images in Figure 1.
page 4, lines 148-149, 154: "Add legend in Figure 1"
Response: Changes were made. However, in the particular case of penetration glands, these were stated through the use of neutral red stain, not through the alum carmine stain whose images were used in figure 1. This detail was added in the material and methods.
page 5, lines 163, 167, 169: "These three sentences are not enough precise"
Response: Changes were made.
page 6, table: "Mean with standard deviation?"
Response: yes, changes were made.
page 12, lines 76-86: "very few information only 1 cercaria"
Response: paragraph was deleted.
page 15, lines 132-134: "Lineage III should be deleted from the manuscript"
Response: It was deleted.
page 15, lines 137-139: "Not for Lineage III"
Reponse: Changes were made.
page 15, lines 146-150: "not enough data"
Response: It was deleted.
page 16, lines 219-220: *sentence highlighted*
Response: sentence was deleted.
page 17, lines 242-244, 248-249: *sentence highlighted*
Response: sentence was deleted.
page 19 (material and methods), lines 376-377: "explain this choice"
Response: information was added.
page 19 : "this information is already given in Figure 6"
Response: that is not correct, in figure 6 only the locations per region were stated, not the different taxa of snails per locality. However, is true that this information was detailed in table 2. Thus, this paragraph was deleted and replaced by a brief sentence.
page 21, table 2: "in bold"
Response: changes were made.
page 21, lines 410-411: "Did the authors checked if any cercariae were shed during the night ?"
Response: No, we didn't.
Reviewer 3 Report
SUMMARY
This is a review of the manuscript entitled “Morphological, behavioral, and molecular characterization of three lineages of avian schistosomes (Digenea: Schistosomatidae) in the native snail Chilina dombeyana (Chilinidae) from southern Chile” (pathogens-1612335) by Oyarzún-Ruiz et al. In their manuscript, the authors characterized the furcocercariae of avian schistosomes from southern Chile sampled in different locations. Snails were sampled, identified, shed for furcocercariae, and dissected. Only Chilina dombeyana was found infected. Morphology (using light and electron microscopy) and behavior of collected furcocercariae were analyzed. Genetic characterization using nuclear (28S) and mitochondrial (COI) markers was performed. These revealed the presence of 3 lineages of avian schistosomes.
This study seems to have been conducted with appropriate methods to characterize furcocercariae. While not being a specialist in parasite taxonomy (at least from morphology), the study sounds robust and clears some of the previous assumptions from the literature. This helps to better understand the diversity and range of parasites in the region. In addition, this study is relevant given the recent outbreak of cercarial dermatitis. Overall the manuscript reads well.
LEGEND
- l.: line
- Fig.: figure
- Tab.: table
MAJOR COMMENTS
The alignment used to build the trees should be provided as a supplementary file.
Comments on the text
-
l. 113: “co-infections between Lineages I and II”: how co-infection was determined. This is not mentioned in the Materials and Methods section.
-
l. 243: “it would correspond to an undescribed”: while the authors took some precautions in this statement, one specimen is clearly not enough, especially given the its state from figure 1. It would be preferable to only state that identification of this specimen was not possible.
-
l. 443: DNA extraction was performed on cercariae and sporocysts. However, as co-infection was observed, how the authors ensured that samples used for the extraction were not carrying several lineages.
MINOR COMMENTS
Comments on the text
-
Title: the title is lengthy and may not get the attention desired. I would suggest to shorten it. Here is a proposition: “Morphological, behavioral, and molecular characterization of avian schistosomes in the snail Chilina dombeyana from southern Chile”.
-
l. 36, 66, 74: the use of the word cranial sounds strange as for a layman it would refers to a skull. Except if this is a common usage in the field, I would suggest to replace this with anterior or apical or similar word.
-
l. 88: “both genes”: the authors should restate what genes they sequenced.
-
l. 91: “lineage 1[12]” should be “lineage 1 [12]” (white space missing).
-
l. 92: “2021[4]” should be “2021 [4]” (white space missing).
-
l. 109: “genus Allobilharzia”: if an article related to the characterization of the genus is available, the authors should add it.
Author Response
REVIEWER 3
line 113: “co-infections between Lineages I and II”: how co-infection was determined. This is not mentioned in the Materials and Methods section.
Response: Details were added in Materials and Methods section. Furcocercariae were collected and separated according to its general morphology because both lineages were clearly different under the stereomicroscope. Thus, we could get specimens of every lineage for staining and molecular characterization.
line 243: “it would correspond to an undescribed”: while the authors took some precautions in this statement, one specimen is clearly not enough, especially given the its state from figure 1. It would be preferable to only state that identification of this specimen was not possible.
Responce: furcocercaria of lineage III was deleted from the manuscript, including images in figure 1.
line 443: "DNA extraction was performed on cercariae and sporocysts. However, as co-infection was observed, how the authors ensured that samples used for the extraction were not carrying several lineages."
Response: Same as indicated above, however, I added the detail that in this particular case only DNA from the furcocercariae of every lineage was extracted separately (see material and methods).
Title: "the title is lengthy and may not get the attention desired. I would suggest to shorten it. Here is a proposition: 'Morphological, behavioral, and molecular characterization of avian schistosomes in the snail Chilina dombeyana from southern Chile' ""
Response: We are agree with the suggestion, although we prefer to keep "(Digenea: Schistosomatidae)" and "native".
lines 36, 66, 74: "the use of the word cranial sounds strange as for a layman it would refers to a skull. Except if this is a common usage in the field, I would suggest to replace this with anterior or apical or similar word."
Response: Changes were made.
line 88: “ 'both genes': the authors should restate what genes they sequenced"
Response: Change was made.
line 91: “lineage 1[12]” should be “lineage 1 [12]” (white space missing)"
Response: Change was made.
line 92: “2021[4]” should be “2021 [4]” (white space missing).
Response: Change was made.
line 109: “genus Allobilharzia”: if an article related to the characterization of the genus is available, the authors should add it.
Response: Article was cited and added to the list of references.
Round 2
Reviewer 2 Report
The authors changed the manuscript as requested; still three changes are needed:
line 25 : delete "with two of these";
line 109 : explain the numbers;
line 110 : "Three" should be replaced by "Two".
Author Response
Round 2, reviewer 2
line 25 : change was made.
line 109 : changes were made
line 110 : change was made.